# An Efficient Fault Detection and Exclusion Method for Ephemeris Monitoring

Yiping Jiang *, Wang Li and Hengwei Zhang

Interdisciplinary Division of Aeronautical and Aviation Engineering, Hong Kong Polytechnic University, Hong Kong SAR, China; 20114095r@connect.polyu.hk (W.L.); hengwei.zhang@connect.polyu.hk (H.Z.)
* Correspondence: yiping.jiang@polyu.edu.hk

**Abstract:** The ephemeris fault needs to be detected and mitigated in the ground-based augmentation system to provide precision approach for the aircraft. In the current fault detection and exclusion (FDE) method, the double-differenced carrier phase (DDCP) observation is used as a test statistic to detect a faulty satellite caused by an ephemeris fault, taking advantage of the residual spatial gradient. However, the current FDE method cannot distinguish whether the fault comes from a reference satellite (RS) or a non-reference satellite (NRS) in DDCP. One way to address this issue is to pre-validate the RS before it can be used to form a DDCP test statistic for detecting ephemeris fault on the NRS. The RS is pre-validated using the previous ephemeris for any newly acquired and re-acquired satellite. This method is developed in detail to present the shortcomings. A more efficient FDE method using multiple hypothesis testing to detect ephemeris fault on both the RS and NRS simultaneously in real time is proposed. Moreover, to facilitate the application in integrity monitoring, the test risks and minimum detectable error are analyzed. The numerical results of the proposed FDE method show an improved performance in detecting ephemeris fault on the RS and a comparable performance on the NRS compared with the current FDE method.

**Keywords:** fault detection and exclusion; spatial gradient monitor; integrity monitoring





## 1. Introduction

The global navigation satellite systems (GNSSs) have been widely applied into various fields and play an important role in civil navigation utility. The ground-based augmentation system (GBAS) is installed in the vicinity of an airport as a local-area differential GNSS system for precision approaches [1,2]. The GBAS uses differential techniques to improve the positioning accuracy as well as provides integrity information to the aircraft during precision approaches. The system failures and anomalies, such as ephemeris fault and ionospheric anomalies, need to be reliably detected [3,4]. Ephemeris faults pose a threat to the aircraft if undetected since the incorrect satellite orbit may be used to compute the aircraft position and result in a large position error. Therefore, it is vital to monitor the ephemeris fault in the GBAS, and the goal of ephemeris monitor is to detect and exclude a faulty satellite before it is used for broadcasting differential corrections to an aircraft.

Satellite ephemeris faults can be categorized into two types, namely, type A with planned satellite maneuvers, and type B without planned satellite maneuvers. The major difference between type A and type B faults is that type B fault can be detected at onset by comparing the ephemeris with an earlier correct ephemeris [5,6]. For GBAS users within the satellite-based augmentation system (SBAS) coverage area, the ephemeris error can be mitigated by the use of SBAS ephemeris correction. However, it is not practical for GBAS users outside the SBAS service area. Therefore, the ephemeris fault mitigation should rely on GBAS capability alone. For CAT I precision approaches, the yesterday-minus-today ephemeris (YE-TE) test has been used in GBAS to monitor the type B ephemeris threat, where newly received ephemeris data are compared with a previously validated ephemeris

or a projection of that ephemeris to the current time [5,7,8]. However, such a monitoring method is unable to protect users against type A ephemeris threat as prior ephemeris data are not used after satellite maneuvers [9–16]. A mitigation method with single range rate monitor is proposed to detect the type A ephemeris threat for CAT I operation [17,18]. For more demanding CAT II/III approaches, the high-precision double-differenced carrier phase (DDCP) observation has been used to monitor ephemeris faults of both type A and type B [9,10,19]. The test statistic using DDCP observation between the reference satellite (RS) and the non-reference satellite (NRS) of two ground receivers is sensitive to the ephemeris fault. The current fault detection and exclusion (FDE) method raises alarm if the test statistic exceeds the threshold. However, the ephemeris fault, if detected, cannot be distinguished whether it occurs on the RS or the NRS.

Although the single fault assumption, that is, only a single satellite fault may exist in the system at any time and the probability of simultaneous ephemeris fault on multiple satellites is negligible, is valid since the ephemeris message are created independently for each satellite and based on the GNSS performance [2,17], it is necessary to distinguish a faulty satellite between the reference satellite (RS) and the non-reference satellite (NRS). To address this issue and to meet the CAT II/III requirement, the RS can be pre-validated beforehand with the previously mentioned YE-TE method or range rate monitor, and the pre-validated RS can then be used to form the test statistic for monitoring the NRS. The pre-validation can be arranged in the time period before the data processing filter reaches the steady state. However, with two fault detection and exclusion (FDE) tests for RS and NRS separately, the computation of probability of false alarm (PFA), probability of misdetection (PMD), and probability of incorrect exclusion (PIE) are shown to be complex to compute. For instance, there is a risk that a faulty reference satellite has not been detected in the pre-validation test and propagates to the second test. Therefore, a new FDE method is proposed to test both RS and NRS in real time, saving the effort of pre-validating the RS beforehand. Compared with the current FDE method, which applies the single hypothesis testing, the proposed FDE method utilizes the multiple hypothesis testing. The single hypothesis testing considers only one test statistic and deals with all test statistics one by one, while the multiple hypothesis testing considers all test statistics at the same time [20]. With combined information from all test statistics, the proposed FDE method can detect the ephemeris fault on both the RS and the NRS simultaneously. To facilitate using the method for integrity monitoring applications, the test risks and the minimum detectable error (MDE) are analyzed.

The rest of this paper is organized as follows: First, the current ephemeris monitor is introduced including both the test statistics and corresponding ambiguity resolution methods. Second, the FDE method with the pre-validation procedure is described in detail with identified issues. Then, a new FDE method is proposed with the outcome test risks, including the PFA, PMD, and the MDE analyzed. Finally, the numerical results regarding the new FDE method are presented.

## 2. Overview of Ephemeris Monitor

The DDCP is adopted as test statistics, where the common errors between two satellites and two reference receivers are canceled. With the precisely surveyed coordinates of the ground receivers, the "observed-minus-computed" L1 DDCP $\varnothing_{ab,1}^{ij}$ with ambiguity resolved is adopted as the test statistic for testing satellite $j$,

$$t_j = \varnothing_{ab,1}^{ij} - \lambda_1 N_{ab,1}^{ij} = I_{ab,1}^{ij} + T_{ab}^{ij} + E_{ab}^{ij} + \varepsilon_{dd_{p1}} \tag{1}$$

where the satellite with the highest elevation angle is denoted as an RS $i$, and the other satellite is denoted as an NRS $j$; subscripts $a$ and $b$ indicate two ground reference stations; $\lambda_1$ is the L1 wavelength; $N_{ab,1}^{ij}$ is the double-differenced ambiguity; the residual atmospheric errors include ionospheric error $I_{ab,1}^{ij}$ and tropospheric error $T_{ab}^{ij}$ after double differencing;

$E_{ab}^{ij}$ is the residual ephemeris error; and $\varepsilon_{dd_{p1}}$ is the residual double-differenced phase error due to multipath and noise.

Under the fault-free hypothesis, $\varepsilon_{dd_{p1}}$ dominates the residual errors and the test statistics can be assumed to follow a normal distribution with a zero mean. Under the faulty hypothesis, a spatial gradient is generated by an ephemeris fault, which is illustrated in Figure 1 and defined in (2),

$$E_{ab}^{ij} = \Delta e_j^T x_{ab} = \frac{\Delta s_j^T \left( I - e_j e_j^T \right) x_{ab}}{r_j} \tag{2}$$

where $e_j$ is the broadcast line-of-sight (LOS) unit vector from a ground station to a satellite $j$; $\Delta e_j^T$ is a difference between the broadcast signal and true LOS unit vector; $x_{ab}$ is the baseline vector between two reference receivers; $\Delta s_j^T$ is the position error vector of a satellite $j$; $I$ is the identity matrix; and $r_j$ is the geometric range from ground receiver $a$ to satellite $j$. According to (2), only the satellite position error orthogonal to the LOS (i.e., $\Delta s_j^T \left( I - e_j e_j^T \right)$) contributes to the range error, also the impact of ephemeris faults on the test statistics is proportional to the length of a ground baseline $x_{ab}$.

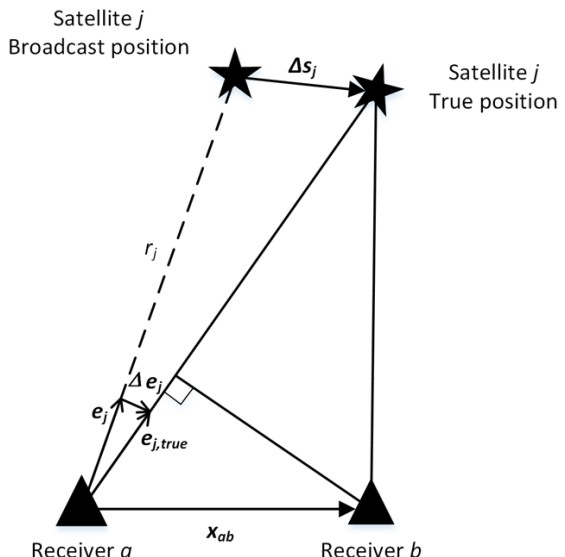

**Figure 1.** Illustration of an ephemeris fault on satellite $j$.

$t_j$ can be used to monitor ephemeris fault only when the unknown ambiguity $N_{ab,1}^{ij}$ is correctly resolved. To isolate the fault with the ambiguity, the statistics for ambiguity resolution need to be geometry-free. The optimal method of resolving $N_{ab,1}^{ij}$ is determined considering the PMD performance and the required number of averaging epochs for newly acquired and re-acquired satellites [11]. First, the wide-lane (WL) ambiguity is estimated by the Melbourne–Wübbena (MW) combination,

$$\frac{\varnothing_{ab,w}^{ij} - R_{ab,n}^{ij}}{\lambda_w} = N_{ab,w}^{ij} + \frac{\varepsilon_{w_p}}{\lambda_w} - \frac{\varepsilon_{n_c}}{\lambda_w} \tag{3}$$

where $\lambda_w = \frac{c}{f_1 - f_5}$ is the WL wavelength at the speed of light $c$; $f$ is the frequency and subscripts "1" and "5" are used to indicate frequencies L1 and L5, respectively $N_{ab,w}^{ij} = N_{ab,1}^{ij} - N_{ab,5}^{ij}$ is the WL ambiguity; and $\varepsilon_{w_p}$ is the residual error in $\varnothing_{ab,w}^{ij}$; $\varnothing_{ab,w}^{ij}$ is the WL combination of phase observations,

$$\varnothing_{ab,w}^{ij} = \frac{f_1 \varnothing_{ab,1}^{ij} - f_5 \varnothing_{ab,5}^{ij}}{f_1 - f_5} \tag{4}$$

where $R_{ab,n}^{ij}$ is the narrow-lane (NL) code combination,

$$R_{ab,n}^{ij} = \frac{f_1 R_{ab,1}^{ij} + f_5 R_{ab,5}^{ij}}{f_1 + f_5} \tag{5}$$

where $R$ is the double-differenced code observation; and $\varepsilon_{n_c}$ is the residual error in $R_{ab,n}^{ij}$.

With only multipath and noise residual errors in MW combination, $\hat{N}_{ab,w}^{ij}$ can be obtained by averaging over a sufficient number of epochs. Then, the L1 DDCP ambiguity is estimated by [11],

$$\frac{\varnothing_{ab,1}^{ij} - \varnothing_{ab,5}^{ij} - \lambda_5 \hat{N}_{ab,w}^{ij}}{\lambda_1 - \lambda_5} = N_{ab,1}^{ij} + \frac{I_{ab,1}^{ij} - I_{ab,5}^{ij}}{\lambda_1 - \lambda_5} + \frac{\varepsilon_{dd_{p1}}}{\lambda_1 - \lambda_5} - \frac{\varepsilon_{dd_{p5}}}{\lambda_1 - \lambda_5} \tag{6}$$

where the residual ionosphere error is assumed to be under nominal state with the assumption of single satellite fault and that two faults cannot exist on a single satellite [2]. The residual multipath and noise become the dominating errors which can also be reduced by averaging among multiple epochs.

With the ambiguity resolved, the tests statistic $t_j$ is highly sensitive to the ephemeris fault as $E_{ab}^{ij}$ exceeds the threshold when the fault occurs. Since $t_j$ is formed by two satellites of the RS and the NRS, it requires one of them to be healthy or fault-free. In the current FDE method, the RS with the highest elevation angle is assumed to be fault-free. This assumption may neglect the risk due to the misdetection of the fault on the RS and incorrect exclusion of the NRS. Therefore, two methods are proposed in the next two sections to address this issue.

## 3. FDE Method with Pre-Validation

The first method is to ensure the RS is fault-free before it can be used in the monitoring of ephemeris fault in the NRS, the pre-validation procedure can be used to detect the ephemeris fault in the RS beforehand. In the pre-validation procedure, the current broadcast ephemeris data in RS are validated by the most recent validated ephemeris data regarding the satellite position or ephemeris parameters [5]. Two test statistics are used, namely $s_i$ for testing the RS $i$ as a difference between two sets of ephemeris data and $t_j$ for testing the NRS. A cascaded test procedure is adopted, where the pre-validation test for an RS is followed by the detection of a spatial gradient for an NRS.

To demonstrate the combined test risks, three cases are considered assuming a single satellite fault. The first case assumes that the RS $i$ is fault-free as well as NRSs with the corresponding results shown in Figure 2. $T_s$ and $T_j$ denote the thresholds for test statistics $s_i$ and $t_j$, respectively. In the pre-validation period, if the RS is detected as faulty with $|s_i| > T_s$, another satellite is selected as the RS to repeat the previous process. The pre-validation is completed when a healthy RS is selected without any alarm generated. It is assumed that at least one healthy satellite can pass the pre-validation process. After the pre-validation process, the maximum $|t_j|$ among all NRSs is compared with the threshold using the healthy RS. The $P_{FA,j}$ is the outcome PFA to alert an NRS $j$.

The second case is comprised of a fault-free RS $i$ and a faulty NRS $k$ with the test procedure shown in Figure 3, where $P_{MD,k}$ is the PMD of the faulty NRS $k$. The third case assumes that an RS $i$ out of the pre-validation test is faulty and all NRSs are fault-free, where $P_{MD,i}$ is the PMD of the faulty RS $i$ as shown in Figure 4. In this case, a faulty RS which passes through the pre-validation test is used for monitoring the NRS. There is a large probability that the faulty RS will cause the test statistic to exceed the threshold and then the NRS is excluded. In this case, the fault-free NRS is incorrectly excluded. However, it is also likely that the test statistic is below the threshold and no alarm is raised. In this case, the RS is misdetected.

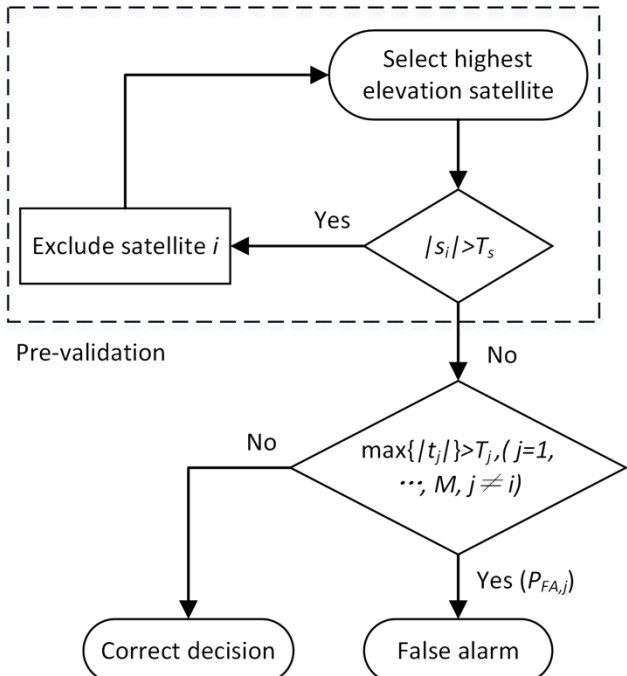

**Figure 2.** Case 1: fault-free RS *i* and fault-free NRSs.

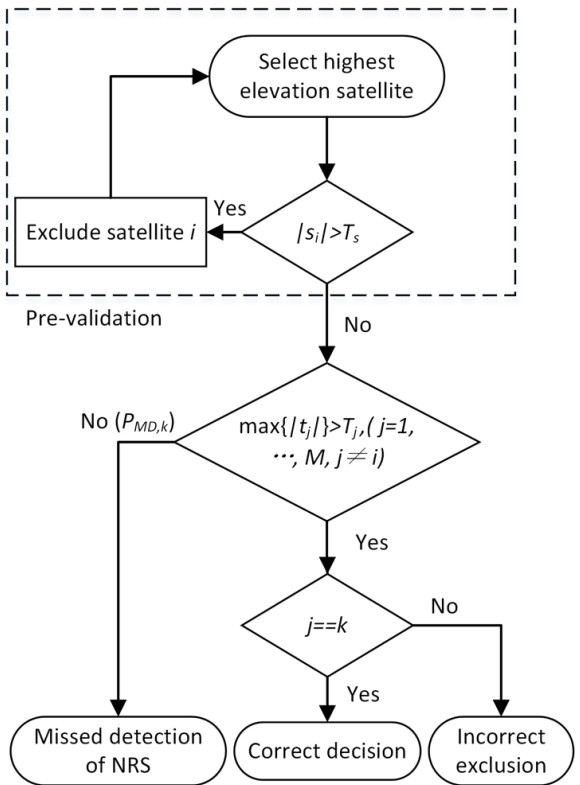

**Figure 3.** Case 2: fault-free RS *i* and faulty NRS *k*.

As shown from Figures 2–4, the test risks including PMD, PFA, and PIE are related with not only the NRS test but also the pre-validation test, making the computation and bounding of test risks complex. Therefore, a more efficient FDE method is proposed in the next section.

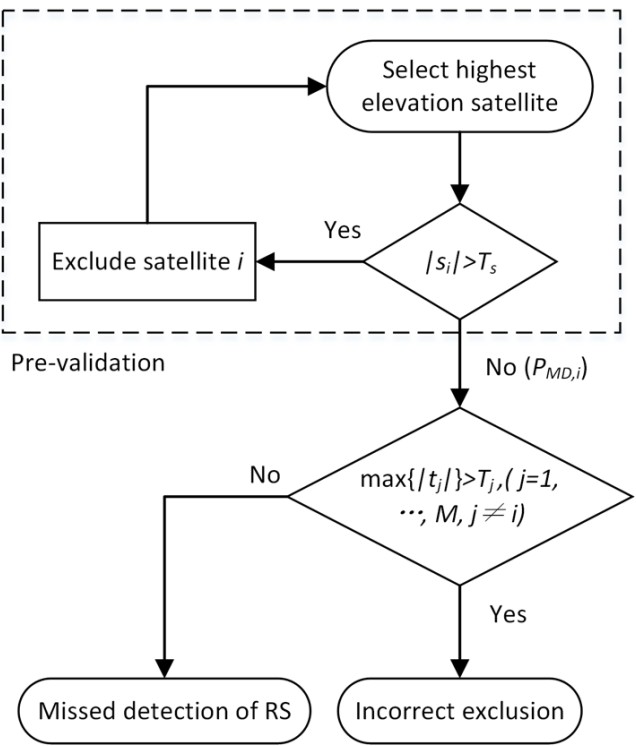

**Figure 4.** Case 3: faulty RS i and fault-free NRSs.

## 4. DD-FDE Method

With the single fault assumption, a new FDE method is proposed to test NRS and RS in a single test, which is referred to as the DD-FDE method. Instead of using the single hypothesis testing in the current FDE method, the DD-FDE method utilizes the multiple hypothesis testing. The pre-validation process is saved and the waiting time for a validated RS is not needed anymore with this method.

### 4.1. Statistical Decisions

To illustrate the DD-FDE method, an example of three visible satellites, including an RS $i$ and NRSs $j$ and $k$, is used. Two test statistics are generated, $t_j$ and $t_k$, for testing two NRSs separately. The test conditions and relevant test risks are illustrated in Table 1. When all test statistics are smaller than the threshold, the fault-free hypothesis is accepted, and when all test statistics exceed the threshold, the faulty RS hypothesis is accepted. A faulty NRS hypothesis is accepted when the following three conditions are satisfied: (1) the corresponding test statistic is maximum among all test statistics; (2) the corresponding test statistic exceeds the threshold; (3) and not all test statistics exceed the threshold.

In Table 1, $H_0$ denotes the fault-free hypothesis, $H_i$, $H_j$, and $H_k$ are faulty hypotheses where indexes $i$, $j$, and $k$ denote the faulty satellites. Comparing the real cases with the test results, different test risks can be obtained, including PFA, PMD, and PIE. To further demonstrate the probabilities of accepting each of the hypotheses, a rectangle space representing the total probability of one is shown in Figure 5. First, the probabilities of $|t_j| > T$, $|t_k| > T$, $|t_j| > |t_k|$, and $|t_j| < |t_k|$ separate the whole space into six subsections of A, B, C, D, E, and F. The areas of subsections are used to indicate the probabilities of accepting each hypothesis in Table 1. In particular, the test result of $H_0$ is represented by the sum of subsections A and F; the test result of $H_i$ is represented by subsections C and D; the test results of $H_j$ and $H_k$ are represented by subsection B and E, respectively.

**Table 1.** The DD-FDE method with an example of three visible satellites.

| Real Case | Test Result | $H_0$ | $H_i$ | $H_j$ | $H_k$ |
|---|---|---|---|---|---|
| | | $\left\|t_j\right\| \leq T \&$ $\|t_k\| \leq T$ | $\left\|t_j\right\| > T \&$ $\|t_k\| > T$ | $\left\|t_j\right\| > T \&$ $\|t_k\| \leq T$ | $\left\|t_j\right\| \leq T \&$ $\|t_k\| > T$ |
| | | A + F | C + D | B | E |
| $H_0$ | | Correct decision | PFA $\alpha_{0i}$ | PFA $\alpha_{0j}$ | PFA $\alpha_{0k}$ |
| $H_i$ | | PMD $\beta_{i0}$ | Correct decision | PIE $\gamma_{ij}$ | PIE $\gamma_{ik}$ |
| $H_j$ | | PMD $\beta_{j0}$ | PIE $\gamma_{ji}$ | Correct decision | PIE $\gamma_{jk}$ |
| $H_k$ | | PMD $\beta_{k0}$ | PIE $\gamma_{ki}$ | PIE $\gamma_{kj}$ | Correct decision |

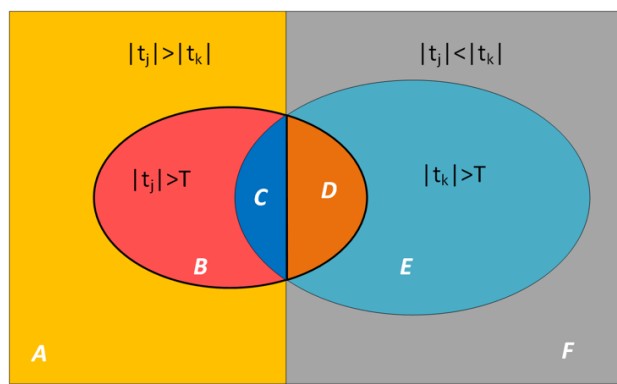

**Figure 5.** Illustration of probabilities of accepting each of the hypotheses.

Numerically, the test risk can be computed with the assumed normal distribution. For instance, the joint distribution of $t_j$ and $t_k$ is provided by,

$$t = \left[t_j, t_k\right]^T \sim N(\mu, \Sigma_t) \tag{7}$$

where $N$ is the two-dimensional normal distribution with a non-central vector $\mu$ and a covariance matrix $\Sigma_t$,

$$\Sigma_t = \begin{bmatrix} \sigma_t{}^2 & \rho_{jk}\sigma_t{}^2 \\ \rho_{jk}\sigma_t{}^2 & \sigma_t{}^2 \end{bmatrix} \tag{8}$$

where $\sigma_t$ is the standard deviation of the test statistic; and $\rho_{jk}$ is the correlation coefficient between two statistics. With multipath and other residual errors from the same RS, there is a correlation between test statistics of different NRSs. The correlation coefficient varies with satellite trajectory and elevation angles. Without loss of generality, the correlation coefficient between two test statistics can be assumed to be arbitrary with a value from a range $(-1, 1)$.

For a general case with $(m + 1)$ visible satellites, the test risks are provided in Table 2. In previous research, it was observed that tropospheric anomalies can also generate a spatial gradient in the DDCP observation [12], which may trigger false alerts in the ephemeris monitor. Considering that the troposphere anomaly is a local phenomenon, dual baselines with enough separation can be used to alleviate this effect [10,11]. Only when both baselines trigger an alarm, is it regarded as an ephemeris fault. Therefore, this test is also generalized with double baseline as shown in Table 3, where superscripts "1" and "2" denote the two baselines. Two baselines are tested separately, and an alert is generated only when test statistics of both baselines meet the criteria in Table 1.

**Table 2.** Results of the DD-FDE method with ($m$ + 1) visible satellites.

| Test Results / Real Case | $H_0$ | $H_i$ | $H_1$ | | $H_k$ |
|---|---|---|---|---|---|
| | $\|t_r\| \leq T$ **For all** $r = 1, 2, \ldots m,$ $r \neq i$ | $\|t_r\| > T$ **For all** $r = 1, 2, \ldots m,$ $r \neq i$ | $\{\mathbf{max}(\|t_r\|) = \|t_1\|\}\&$ $\{\|t_1\| > T\}\&$ $\{\|t_r\| \leq T$ **For one or more** $r = 1\ldots m, r \neq i\}$ | $\ldots$ | $\mathbf{max}(\|t_r\|) = \|t_k\|\}\&$ $\{\|t_k\| > T\}\&$ $\{\|t_r\| \leq T$ **For one or more** $r = 1\ldots m, r \neq i\}$ |
| $H_0$ | Correct decision | PFA $\alpha_{0i}$ | PFA $\alpha_{01}$ | $\ldots$ | PFA $\alpha_{0m}$ |
| $H_i$ | PMD $\beta_{i0}$ | Correct decision | PIE $\gamma_{i1}$ | $\ldots$ | PIE $\gamma_{im}$ |
| $H_1$ | PMD $\beta_{10}$ | PIE $\gamma_{1i}$ | Correct decision | $\ldots$ | PIE $\gamma_{1m}$ |
| $\vdots$ | $\vdots$ | $\vdots$ | $\vdots$ | $\ddots$ | $\vdots$ |
| $H_m$ | PMD $\beta_{m0}$ | PIE $\gamma_{mi}$ | PIE $\gamma_{m1}$ | $\ldots$ | Correct decision |

**Table 3.** Results of the DD-FDE method with double baseline.

| | $H_0$ | $H_i$ | $H_1$ | | $H_k$ |
|---|---|---|---|---|---|
| $H_0$ | Correct decision | PFA $\alpha_{0i}^1 \cap \alpha_{0i}^2$ | PFA $\alpha_{01}^1 \cap \alpha_{01}^2$ | $\ldots$ | PFA $\alpha_{0m}^1 \cap \alpha_{0m}^2$ |
| $H_i$ | PMD $\beta_{i0}^1 \cup \beta_{i0}^2$ | Correct decision | PIE $\gamma_{i1}^1 \cap \gamma_{i1}^2$ | $\ldots$ | PIE $\gamma_{im}^1 \cap \gamma_{im}^2$ |
| $H_1$ | PMD $\beta_{10}^1 \cup \beta_{10}^2$ | PIE $\gamma_{1i}^1 \cap \gamma_{1i}^2$ | Correct decision | $\ldots$ | PIE $\gamma_{1m}^1 \cap \gamma_{1m}^2$ |
| $\vdots$ | $\vdots$ | $\vdots$ | $\vdots$ | $\ddots$ | $\vdots$ |
| $H_m$ | PMD $\beta_{n0}^1 \cup \beta_{n0}^2$ | PIE $\gamma_{ni}^1 \cap \gamma_{ni}^2$ | PIE $\gamma_{m1}^1 \cap \gamma_{m1}^2$ | $\ldots$ | Correct decision |

*4.2. Test Risks and MDE Derivation*

As shown in the last section, the multi-variant normal distribution with a varying correlation coefficient significantly increases the complexity for calculation of test risks. The following demonstrates that pre-defined values can bound the test risks with a single-variant normal distribution for any correlation coefficient. An example of a single baseline with three visible satellites is illustrated below. First, $\alpha_0$ and $\beta_0$ are defined by the single-variant normal distribution,

$$\alpha_0 = P(|t_j| > T | H_0) \tag{9}$$

$$\beta_0 = P(|t_j| \leq T | H_j) \tag{10}$$

where $T$ is the threshold that is used in the following risk definitions. The test risks in Table 1 can be bounded by $\alpha_0$ and $\beta_0$ as follows,

$$\alpha_{0i} = P(|t_j| > T \cap |t_k| > T | H_0) \leq \alpha_0 \tag{11}$$

$$\alpha_{0j} = P(|t_j| > T \cap |t_k| \leq T | H_0) \leq \alpha_0 \tag{12}$$

$$\beta_{i0} = P(|t_j| \leq T \cap |t_k| \leq T | H_i) \leq \beta_0 \tag{13}$$

$$\beta_{j0} = P(|t_j| \leq T \cap |t_k| \leq T | H_j) \leq \beta_0 \tag{14}$$

$$\gamma_{ji} = P\left(|t_j| > T \cap |t_k| > T|H_j\right) \le \alpha_0 \tag{15}$$

$$\gamma_{kj} = P\left(|t_j| > T \cap |t_k| \le T|H_k\right) \le \alpha_0 \tag{16}$$

where an arbitrary correlation coefficient between $t_j$ and $t_k$ is adopted. Similar conclusions can be derived for more visible satellites. For the case of double baseline, it holds that,

$$\left(\alpha_{0i}^1 \cap \alpha_{0i}^2\right) < \alpha_0 \tag{17}$$

$$\left(\beta_{i0}^1 \cup \beta_{i0}^2\right) < 1 - (1 - \beta_0)^2 \tag{18}$$

$$\left(\gamma_{ih}^1 \cap \gamma_{ih}^2\right) < \alpha_0 \tag{19}$$

where an arbitrary correlation coefficient between two baselines is adopted. It is concluded that the PFA and PIE with double baseline can be bounded by $\alpha_0$, and PMD with double baseline is bounded by $1 - (1 - \beta_0)^2$. Therefore, incorporating a second baseline to protect users against false alarms results in an increase in PMD with $\left(1 - (1 - \beta_0)^2\right) \ge \beta_0$.

The purpose of bounding test risks with $\alpha_0$ and $\beta_0$ is to simplify the proceeding calculations of thresholds and protection levels. Particularly, with $\alpha_0$ as the required PFA value, $T$ can be derived directly from (9) to avoid the complexity of the multi-variate distribution and the varying correlation coefficient. In this way, the resulted PFAs with the DD-FDE method can be guaranteed to be lower than the required value with (17). The significance of bounding PMD is demonstrated in the next section for calculating the protection level.

The undetected ephemeris error needs to be bounded by VPL in the vertical position domain [13]. An ephemeris $p$ value is broadcast to users for calculation of VPL [1,14],

$$P_j = \frac{MDE}{x_{ab}} = \frac{u_j}{x_{ab}} \tag{20}$$

where MDE denotes the MDE in the differential range domain with given PMD; MDE is equal to the non-centrality bias in the test statistic, which is denoted as $u_j$ under hypothesis $H_j$; and $x_{ab}$ is the baseline length of the ground receivers. The ephemeris vertical protection level $VPL_e^j$ under hypothesis $H_j$ is [5,14],

$$VPL_e^j = |S_{3,j}|P_jb + K_{md}\sqrt{\sum_{k=1}^{N} S_{vert,k}^2 \sigma_k^2} \tag{21}$$

where $S$ is the projection matrix from the least-squares estimation; $b$ is the distance between an aircraft and ground stations; $N$ is the total number of visible satellites; $\sigma_k$ is the standard deviation of the differential range error for a satellite $k$; and $K_{md}$ is the broadcast $K$-value. The maximum VPL among all hypotheses is the final VPL. With the previous FDE method, the VPL under a hypothesis of a faulty RS is not derived. With the proposed DD-FDE method, VPL can be obtained for all satellites.

Since VPL value is a function of $P_j$, which is determined by MDE, it is necessary to analyze MDE for the proposed FDE method and compare it with MDE of the previous FDE method. The current FDE method uses single hypothesis testing, and $P_{md}$ is defined as the probability that $t_j$ is inside the threshold region under $H_j$, which has a Gaussian distribution with mean of $u$ and standard deviation of $\sigma_t$, i.e., $P_{md}=P\left(|t_j| \le T|H_j\right)$. With $\beta_0$ allocated to $P_{md}$ as indicated by (10), the non-centrality bias is obtained by,

$$u = T + Q^{-1}(1 - \beta_0)\sigma_t \tag{22}$$

where $Q^{-1}$ as the inverse of the standard normal inverse cumulative distribution function. $T$ is computed by $Q^{-1}\left(1 - P_{fa}\right)\sigma_t$ from (9).

On the other hand, the proposed FDE method uses multiple hypothesis testing, and the expression of $P_{md}$ is indicated by the sum of $\beta_{i0}$ and $\beta_{r0}(r = 1, \cdots, m - 1)$ considering $m$ satellites. Assuming $\beta_0$ is equally allocated to all satellites, $P_{md}$ for the RS and NRS are $\beta_{i0} = \beta_{r0} = \frac{\beta_0}{m}(r = 1, \ldots, m - 1)$. With $m$ satellites in view, $P_{md}$ for the NRS $j$ is defined as the probability that all the test statistics are inside the threshold region under $H_j$. In this case, $t_j$ follows a Gaussian distribution with a mean of $u_j$ and standard deviation of $\sigma_t$, while $t_i$ and $t_r(r = 1, \ldots, m - 1, \text{and} r \neq j)$ follow a Gaussian distribution with a mean of zero. Then, $P_{md}$ for the NRS $j$ can be written as,

$$P\left(\left|t_j\right| \leq T \cap |t_i| \leq T \cap |t_r| \leq T\Big|H_j\right) = \beta_{j0} = \frac{\beta_0}{m}, (r = 1, \ldots, m - 1, \text{and} r \neq j) \tag{23}$$

where the probabilities of $|t_i| \leq T$ and $|t_r| \leq T$ are approximately to 1. Therefore, (23) can be approximated as,

$$P\left(\left|t_j\right| \leq T\Big|H_j\right) \approx \frac{\beta_0}{m} \tag{24}$$

and $u_j$ can be obtained by,

$$u_j \approx T + Q^{-1}\left(1 - \frac{\beta_0}{m}\right)\sigma_t \tag{25}$$

which is larger than $u$ compared with (22). In other words, the non-centrality bias for the NRS in the new FDE method is larger than that in the current FDE method.

For the RS $i$ in the new FDE method, $P_{md}$ is defined as the probability that all the test statistics are inside the threshold region under $H_i$. In this case, all test statistics follow a Gaussian distribution with mean of $u_i$ and standard deviation of $\sigma_t$ since the bias $u_i$ in the RS is propagated to other NRS. Then, $P_{md}$ for the RS $i$ can be written as,

$$P(|t_i| \leq T \cap |t_r| \leq T|H_i) = \beta_{i0} = \frac{\beta_0}{m}, (r = 1, \ldots, m - 1) \tag{26}$$

where the probabilities of $|t_i| \leq T$ and $|t_r| \leq T$ are equal. Considering the independence between all test statistics, (26) can be rewritten as,

$$P(|t_i| \leq T|H_i) = \left(\frac{\beta_0}{m}\right)^{\frac{1}{m}}, (r = 1, \ldots, m - 1) \tag{27}$$

and $u_i$ can be obtained by,

$$u_i = T + Q^{-1}\left(1 - \left(\frac{\beta_0}{m}\right)^{\frac{1}{m}}\right)\sigma_t \tag{28}$$

which is smaller than $u$ compared with (22) since $\left(\frac{\beta_0}{m}\right)^{\frac{1}{m}}$ is larger than $\beta_0$. In other words, the non-centrality bias for the NRS in the new FDE method is larger than that in the current FDE method.

For the case of double baselines, $P_{md}$ for the NRS is defined as the probability that either of the double baseline misdetect the ephemeris fault. Based on the $P_{md}$ analysis for the NRS of the single baseline, $P_{md}$ for the NRS under double baselines is,

$$P\left(\left|t_j^1\right| \leq T\Big|H_j\right) \cup P\left(\left|t_j^2\right| \leq T\Big|H_j\right) \approx \frac{\beta_0}{m} \tag{29}$$

with the risk overbounding result derived in (18), the non-centrality of the NRS for the double baselines can be obtained by,

$$u_j' \approx T + Q^{-1}\left(\sqrt{1 - \frac{\beta_0}{m}}\right)\sigma_t \tag{30}$$

Similarly, the non-centrality of the RS for the double baselines can be obtained by substituting $\sqrt{1 - \frac{\beta_0}{m}}$ into (28),

$$u_i' = T + Q^{-1}\left( \left(1 - \frac{\beta_0}{m}\right)^{\frac{1}{2m}} \right)\sigma_t \tag{31}$$

where the additional $P_{md}$ is introduced by the second baseline, and $u_i'$ as well as $u_j'$ are larger than $u_i$ and $u_j$, respectively.

## 5. Numerical Results of DD-FDE Method

With $\sigma$ bounded as 0.6 cm [9], the numerical results of PFA under the fault-free case with $\mu = [0, 0]^T$ are obtained in Table 4 as a function of $\rho_{jk}$. It can be observed that $\rho_{jk}$ has a slight influence on $\alpha_{0j}$ and $\alpha_{0k}$, while $\alpha_{0i}$ increases obviously with the increase in $\rho_{jk}$. In other words, PFA of an alerted RS is easier to be triggered when test statistics are more correlated, while the PFA of an alerted NRS is not obviously influenced by the correlation among test statistics.

**Table 4.** Test risk results for the fault-free case with three visible satellites.

| $\rho_{jk}$ | $\alpha_{0i}$ | $\alpha_{0j}$ | $\alpha_{0k}$ |
|---|---|---|---|
| 0.9 | $1.8 \times 10^{-9}$ | $8.2 \times 10^{-9}$ | $8.2 \times 10^{-9}$ |
| 0.6 | $3.3 \times 10^{-11}$ | $1.0 \times 10^{-8}$ | $1.0 \times 10^{-8}$ |
| 0.3 | $1.7 \times 10^{-13}$ | $1.0 \times 10^{-8}$ | $1.0 \times 10^{-8}$ |
| 0 | $7.5 \times 10^{-17}$ | $1.0 \times 10^{-8}$ | $1.0 \times 10^{-8}$ |

For the faulty RS case, an example with $\mu = [0.065, 0.065]^T$ is used to calculate the risk values. As shown in Table 5, $\beta_{i0}$ increases obviously when $\rho_{jk}$ increases, while other probabilities remain relatively stable. In other words, the PMD of a faulty RS is easier with more correlated test statistics, while the PIE of a fault-free NRS is not obviously affected by the correlation level among test statistics.

**Table 5.** Test risk results for the faulty RS case with three visible satellites.

| $\rho_{jk}$ | $\beta_{i0}$ | $\gamma_{ij}$ | $\gamma_{ik}$ |
|---|---|---|---|
| 0.9 | $3.7 \times 10^{-8}$ | $1.3 \times 10^{-7}$ | $1.3 \times 10^{-7}$ |
| 0.6 | $1.4 \times 10^{-9}$ | $1.7 \times 10^{-7}$ | $1.7 \times 10^{-7}$ |
| 0.3 | $1.9 \times 10^{-11}$ | $1.7 \times 10^{-7}$ | $1.7 \times 10^{-7}$ |
| 0 | $2.8 \times 10^{-14}$ | $1.7 \times 10^{-7}$ | $1.7 \times 10^{-7}$ |

For the faulty NRS case, an example with $\mu = [0.065, 0]^T$ is used to calculate the risk values. As shown in Table 6, $\gamma_{ji}$ and $\beta_{j0}$ are relatively stable when $\rho_{jk}$ varies, while $\gamma_{jk}$ increases obviously with the increase in $\rho_{jk}$. In other words, a higher correlation between test statistics makes it harder to distinguish the fault among different NRSs.

**Table 6.** Test risk results for the faulty NRS case with three visible satellites.

| $\rho_{jk}$ | $\beta_{j0}$ | $\gamma_{ji}$ | $\gamma_{jk}$ |
|---|---|---|---|
| 0.9 | $1.6 \times 10^{-7}$ | $6.7 \times 10^{-9}$ | $3.3 \times 10^{-9}$ |
| 0.6 | $1.7 \times 10^{-7}$ | $1.0 \times 10^{-8}$ | $1.3 \times 10^{-10}$ |
| 0.3 | $1.7 \times 10^{-7}$ | $1.0 \times 10^{-8}$ | $1.2 \times 10^{-12}$ |
| 0 | $1.7 \times 10^{-7}$ | $1.0 \times 10^{-8}$ | $1.7 \times 10^{-15}$ |

The numerical results for the fault-free case with eight visible satellites are shown in Table 7, where $\mu = [0, 0, 0, 0, 0, 0, 0]^T$, and $\rho_{jk}$ between any two test statistics is assumed to be the same. Compared with the results in Table 2, the PFA results in Table 7 decrease with

the increase in the number of visible satellites. For the double baseline, numerical results are shown in Table 8, where PFA decreases with the increases in the number of baselines.

**Table 7.** Test risk results for the fault-free case with eight visible satellites.

| $\rho_{jk}$ | $\alpha_{0i}$ | $\alpha_{01}$ | $\cdots$ | $\alpha_{07}$ |
|---|---|---|---|---|
| 0.9 | $1.6 \times 10^{-7}$ | $6.7 \times 10^{-9}$ | $\cdots$ | $3.1 \times 10^{-14}$ |
| 0.6 | $1.7 \times 10^{-7}$ | $1.0 \times 10^{-8}$ | $\cdots$ | $2.4 \times 10^{-9}$ |
| 0.3 | $1.7 \times 10^{-7}$ | $1.0 \times 10^{-8}$ | $\cdots$ | $8.6 \times 10^{-9}$ |
| 0 | $1.7 \times 10^{-7}$ | $1.0 \times 10^{-8}$ | $\cdots$ | $1.0 \times 10^{-8}$ |

**Table 8.** Test risk results for the fault-free case with eight visible satellites and a double baseline.

| $\rho_{jk}$ | $\alpha_{0i}^1 \cap \alpha_{0i}^2$ | $\alpha_{01}^1 \cap \alpha_{01}^2$ | $\cdots$ | $\alpha_{07}^1 \cap \alpha_{07}^2$ |
|---|---|---|---|---|
| 0.9 | $1.4 \times 10^{-16}$ | $9.6 \times 10^{-28}$ | $\cdots$ | $9.6 \times 10^{-28}$ |
| 0.6 | $3.0 \times 10^{-25}$ | $5.8 \times 10^{-18}$ | $\cdots$ | $5.8 \times 10^{-18}$ |
| 0.3 | $1.7 \times 10^{-42}$ | $7.4 \times 10^{-17}$ | $\cdots$ | $7.4 \times 10^{-17}$ |
| 0 | $1.4 \times 10^{-112}$ | $1.0 \times 10^{-16}$ | $\cdots$ | $1.0 \times 10^{-16}$ |

Similar results can be obtained for PMD and PIE, where both PMD and PIE decrease with the number of visible satellites increases, while PMD increases and PIE decreases with the number of baselines increases.

The numerical results of the non-centralities are illustrated, further considering the correlation between the test statistics. Assuming that the allocated $P_{md}$ for ephemeris monitor is $5 \times 10^{-7}$ with three satellites, $1.67 \times 10^{-7}$ is allocated for each hypothesis. The PFA allocated to the ephemeris monitor is $10^{-8}$ for GBAS CAT II/III approaches [15]. With $\sigma_t = 0.6$ cm, $T$ is obtained as 3.5 cm. $u_i$ and $u_j$ are derived using the MATLAB function "fsolve". Figure 6 shows the $u_i$ and $u_j$ results as a function of correlation coefficient $\rho$ with three visible satellites under single baseline configuration.

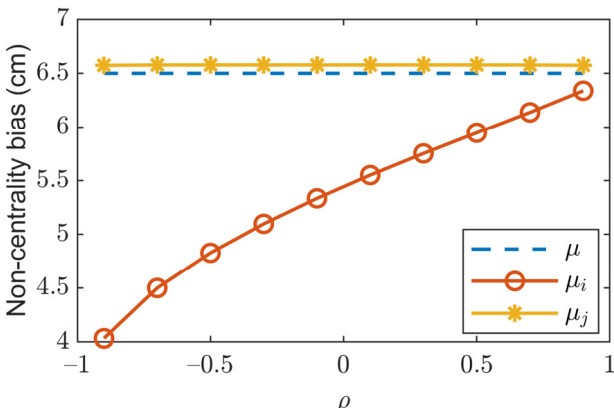

**Figure 6.** Non-centrality bias results for different $\rho$ with three visible satellites in the single baseline case.

As shown in Figure 6, $u_i$ is smaller than $u$ while $u_j$ is larger than $u$, which is consistent with the above analysis. In addition, with the increase in $\rho$, $u_i$ increases sharply while $u_j$ is relatively stable. That is, the MDE of the RS can be affected by the correlation between the test statistics since the fault in the RS will be propagated into other satellites when forming the test statistics. In contrast, the MDE of the NRS is not affected by the correlation between the test statistics because the fault in the NRS only remains within itself. Figure 7 shows the $u_i$ and $u_j$ results as a function the number of visible satellites under the single baseline configuration.

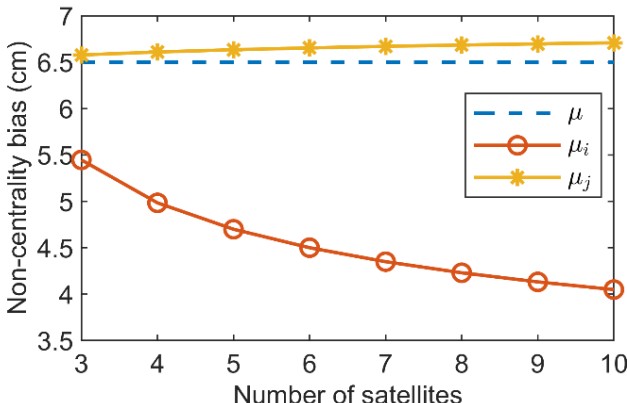

**Figure 7.** Non-centrality bias results for different number of satellites in the single baseline case.

The results in Figure 7 are obtained assuming that the test statistics are independent of each other, i.e., $\rho = 0$. With the number of satellite increases, $u_i$ decreases sharply while $u_j$ has a slight increase. This can also be interpreted by (25) and (28) that a larger number of satellite m will result in a smaller $u_i$ and a larger $u_j$. In addition, the $u_i$ and $u_j$ results for a different number of visible satellites under the double baselines configuration are shown and compared with the single baseline configuration in Figure 8.

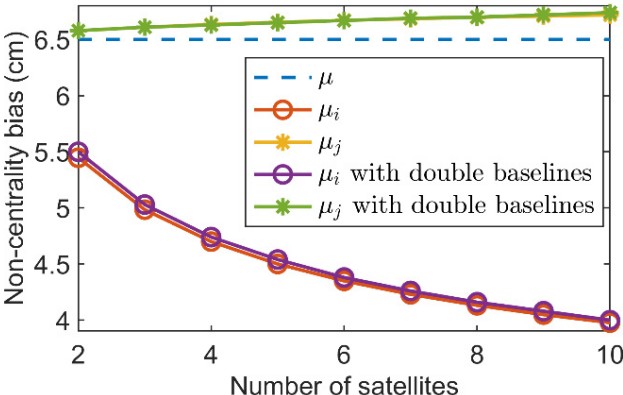

**Figure 8.** Non-centrality bias results for different number of satellites in the double baseline case.

The results in Figure 8 are obtained assuming that the test statistics are independent of each other, i.e., $\rho = 0$. As shown in Figure 8, due to the additional misdetection risk induced by the second baseline, both $u_i$ and $u_j$ in the double baseline case are slightly larger than that in the single baseline case.

In summary, the numerical results of non-centrality of the new FDE method are consistent with the analysis with the derived equations of (25), (28), (30), and (31). Compared with the current FDE method, the non-centrality or MDE for the new FDE method varies with the RS, the NRS, the different correlation coefficient, and the number of visible satellites. In particular, the MDE for the RS is smaller in the new FDE method and decreases with a smaller correlation coefficient and a larger number of satellites, thereby resulting in a smaller VPL. As for the NRS, the MDE is larger in the new FDE method and increases with the number of satellites, thereby resulting in a larger VPL. Furthermore, introducing a double baseline can reduce false alarms induced by the tropospheric turbulence, while the integrity risk is sacrificed with enlarged MDE.

## 6. Discussion and Conclusions

The current FDE method uses DDCP as test statistics but it cannot distinguish whether the fault comes from the RS or the NRS. Although adding pre-validation for the RS can

address this issue, the resultant test risks are difficult to be accounted for. Therefore, the DD-FDE method for ephemeris monitor is proposed in this paper, where faulty RS and NRS can be simultaneously detected by using a multiple alternative hypothesis testing technique. The outcome test risks of PFA, PMD, and PIE are derived and these risks can be bounded by pre-defined values with the single-variant normal distribution. Based on the test risk analysis, the non-centrality or MDE of the DD-FDE method is analyzed. The numerical test results are consistent with the test risk and MDE analysis. In addition, compared with the current FDE method, the MDE for the RS is smaller and decreases sharply with the correlation coefficient between test statistics decreases and the number of visible satellites increases. On the contrary, the MDE for the NRS of the DD-FDE method is slightly larger than that of the current FDE method, and it increases slightly with the number of visible satellite increases. This indicates that the DD-FDE method has an improved ability of detecting ephemeris fault on the RS and a comparable ability on the NRS compared with the current FDE method. Considering the advantages of detecting the ephemeris fault at the RS and NRS simultaneously and the comparable detecting capability, the DD-FDE method shows promising benefits in GBAS application. However, this method needs additional analysis and testing to prove its efficacy. In addition, this method is based on the single-fault assumption, future work will consider ephemeris fault induced by noise in a communication channel and a condition where multiple ephemeris faults occur at the same time.

**Author Contributions:** Conceptualization, Y.J.; methodology, Y.J., W.L. and H.Z.; validation, Y.J., W.L. and H.Z.; writing—original draft preparation, Y.J.; writing—review and editing, W.L.; supervision, Y.J.; project administration, Y.J.; funding acquisition, Y.J. All authors have read and agreed to the published version of the manuscript.

**Funding:** This research was funded by the National Science Foundation of China (no. 42004029), the University Grants Committee/Research Grants Council (no. 25202520).

**Data Availability Statement:** Data sharing is not applicable to this article as no new data were created or analyzed in this study.

**Conflicts of Interest:** The authors declare no conflict of interest.

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
