# Peer review of "An Efficient Fault Detection and Exclusion Method for Ephemeris Monitoring"

_remotesensing, doi:10.3390/rs15133259_

Round 1

Reviewer 1 Report

An Efficient Fault Detection and Exclusion Method for Ephemeris Monitoring based on DDCP is developed and tested in this paper. It will draw lots of research interest in the GBAS area. Some minor revisions can be made for the following points:

1.     Fig. 3. Incorrect exclusion is the false alarm.

2.     Fig. 4. If the faulty RS is missed, all the test statistics should be more than the threshold.

3.     Eq. (9) and (12): what is &?

4.     Numerical result of VPLe would be better to show the performance of proposed method.

Reviewer 3 Report

Authors propose a system for detection of a satellite, having ephemeris - related issues and, therefore, providing false data to ground stations.

I suggest to authors to make changes:

1. In title - having more precise contribution emphasized regarding the proposed method and relating the content with the title, since the content focuses on satellites and no satellites were mentioned in title.

2. In manuscript content - a) since satellites are used for GPS systems, triangulation of satellites is used as a method for general positioning, so this should be mentioned in text (at least in discussion section), to have this research put into context of the realistic situation of satellite usage. b) what is the impact of environment to results of satellite broadcasting, i.e. what influences on noise in communication channel? This should be addressed at least in discussion section and future work part of conclusion.

3. In introduction - authors should, in separate paragraphs, make careful presentation of research motivation, basic terms introduction, short overview of existing research results, emphasis on contribution of this manuscript compared to previous research from literature, and finally, the structure of the rest of the paper. All these aspects are currently provided in certain level, but should be clearly separated in paragraphs and provided with more details.

4. In manuscript structure - authors should make clear distinction between background (having explained basic terms), related work (having previous research on the same or similar problems solution i.e. literature review provided), the proposed method (the contribution), the research methodology (having hypotheses and experimental setup clearly explained - research sample, method of sample collection, experimental environment), research results, discussion and conclusion. Currently, in each of sections (2-6) there is a mix of proposed methods and results, but it is not clear what is research methodology used for such results. Therefore, it is necessary to have separate sections after introduction and they should be entitled and provided with this order: Background, Related work, Proposed Method, Research Methodology, Results, Discussion, Conclusions.    

5. In conclusion - having assumed that only one satellite could be faulty (mentioned in abstract), authors should provide directions for future research regarding this restriction, i.e. having mentioned how current results could relate to more general situations, which are closer to reality. Conclusion usually consists of brief summary of obtained results, comparing to previous research, in aim to emphasize the contribution, as well as to address restrictions of this manuscript and propose future work. It is necessary to provide sentences that could have these aspects covered.

Reviewer 4 Report

1. The manuscript's motivations should be further highlighted in the manuscript, e.g., what problems did the previous works exist? How to solve these problems? The authors may consider analyzing the problems of the previous works and how to address these problems with the proposed method. Please explain that. 2. The research gaps in the abstract and introduction should be clearly expressed. Please rewrite this part. 3. The authors must clearly explain the difference(s) between the proposed method and similar works in the introduction. The authors should further highlight the manuscript's innovations and contributions.
4. The literature review is poor in this paper. You must review all significant similar works that have been done. I hope that the authors can add some new references in order to improve the reviews and the connection with the literatures.
https://doi.org/10.1109/TR.2022.3180273;
http://dx.doi.org/10.1109/TCSS.2022.3152091;
http://dx.doi.org/10.1016/j.oceaneng.2022.113424;
http://dx.doi.org/10.1016/j.marstruc.2022.10333 and so on.
5. The main contributions of this paper should be further summarized and clearly demonstrated.
6. 
The authors are requested to correct all spelling mistakes. The authors are requested to correct all spelling mistakes.

Round 2

Reviewer 4 Report

This paper can be accepted now.

Minor editing of English language required